# The Role of Flies as Pollinators of Horticultural Crops: An Australian Case Study with Worldwide Relevance

**DOI:** 10.3390/insects11060341

**Published:** 2020-06-02

**Authors:** David F Cook, Sasha C Voss, Jonathan T D Finch, Romina C Rader, James M Cook, Cameron J Spurr

**Affiliations:** 1Department of Primary Industries and Regional Development, 3 Baron-Hay Court, South Perth, WA 6151, Australia; 2School of Biological Sciences, The University of Western Australia, 35 Stirling Highway, Crawley, WA 6009, Australia; sasha.voss@uwa.edu.au; 3Plants Animals and Interactions, Western Sydney University, Locked Bag 1797, Penrith, NSW 2751, Australia; j.finch@westernsydney.edu.au (J.T.D.F.); james.cook@westernsydney.edu.au (J.M.C.); 4School of Environmental and Rural Science, University of New England, Madgewick Drive, Armidale, NSW 2351, Australia; rrader@une.edu.au; 5SeedPurity Pty Ltd., 2 Derwent Avenue, Margate, Tasmania 7054, Australia; cspurr@seedpurity.com

**Keywords:** pollination, Diptera, Syrphidae, Rhiniidae, fly, flower visitation, horticulture, foraging, managed pollinators, life history

## Abstract

Australian horticulture relies heavily on the introduced managed honey bee, *Apis mellifera* Linnaeus 1758 (Hymenoptera: Apidae), to pollinate crops. Given the risks associated with reliance upon a single species, it would be prudent to identify other taxa that could be managed to provide crop pollination services. We reviewed the literature relating to the distribution, efficiency and management potential of a number of flies (Diptera) known to visit pollinator-dependent crops in Australia and worldwide. Applying this information, we identified the taxa most suitable to play a greater role as managed pollinators in Australian crops. Of the taxa reviewed, flower visitation by representatives from the dipteran families Calliphoridae, Rhiniidae and Syrphidae was frequently reported in the literature. While data available are limited, there was clear evidence of pollination by these flies in a range of crops. A review of fly morphology, foraging behaviour and physiology revealed considerable potential for their development as managed pollinators, either alone or to augment honey bee services. Considering existing pollination evidence, along with the distribution, morphology, behaviour and life history traits of introduced and endemic species, 11 calliphorid, two rhiniid and seven syrphid species were identified as candidates with high potential for use in Australian managed pollination services. Research directions for the comprehensive assessment of the pollination abilities of the identified taxa to facilitate their development as a pollination service are described. This triage approach to identifying species with high potential to become significant managed pollinators at local or regional levels is clearly widely applicable to other countries and taxa.

## 1. Introduction

Globally, approximately 80% of crop plants are dependent on, or have their yield enhanced by, insect pollinators [1]. The annual gross economic value of crops requiring pollination services is estimated to exceed US $ 780 billion worldwide [2] and approximately US $ 6 billion in Australia [3]. The majority of crops pollinated by insects worldwide are serviced by bees (92%), while flies are the second most important visitors of flowers [4,5,6,7]. Owing to various stresses on honey bees worldwide, along with increasing evidence of wild pollinator decline and concerns about inadequate pollination [8,9], modern crop production will become increasingly dependent on managed pollinators to improve yields. Identifying candidate pollinators and developing management strategies for their commercial use is now crucial to future proof pollinator-dependent crop production systems both globally and in Australian horticulture.

Flies are one of the most diverse groups in the world and are present in nearly all habitats and biomes [5], but have been studied far less than bees. Flies can be as efficient as, or better than, bees for pollinating some crops [10,11,12,13], and are often responsible for transporting high pollen loads in both natural and modified systems [12,14]. Many of these “unmanaged pollinators” and/or “exotic flies” can thus provide consistent pollination to certain crops [13,15,16,17].

As pollinators, flies are likely to represent a good alternative or supplemental option to bees because different species are present all year round and frequently visit flowers to feed on nectar and/or pollen to support key biological functions including flight and reproduction [18]. Being hairy, they also pick up and move pollen from a wide variety of flowers [19,20]. Fly taxa are highly variable with respect to body size, which can be matched to the floral morphology of a target crop for pollination-either by species selection, or within species by manipulating nutrition of the larval stage [21]. In addition, some fly taxa are already easily mass-reared with reasonably low inputs, manageable health and safety requirements and present negligible risk of disease transmission to existing wild pollinators when reared under screened colony conditions. Furthermore, they do not sting farm workers!

Here, we review the literature to identify the fly taxa with the greatest potential for commercial development to supplement bees in pollination services within Australian horticulture. The identification of potential fly pollinators offers further value in directing research efforts around management techniques that facilitate wild pollinator populations, in addition to the development of managed pollination services. Numerous studies have reported fly taxa in association with flower visitation, but less is known regarding the pollination efficiency and effectiveness of individual species across different crops that are reliant upon or enhanced by insect pollination. While almost all horticultural crops are exotic to Australia (macadamia is one such exception), such crops likely represent an additional source of nutrition to both native and exotic fly species that are known generalist foragers. This review aims to examine the few existing reports of fly pollination against the characteristics associated with pollination success, in order to identify the fly taxa most likely to possess the relevant morphology, foraging behaviour and life history traits suitable for pollinating key target crops within Australia. While reviewed in an Australian context, it is with the broader intention that this approach guides the process of identification of species of high potential for development as managed pollinators in other cropping systems and regions of the world. Regionally-specific assessment of suitable taxa is a necessary process to avoid unwanted/pest interactions with other pollinators and locally occurring biodiversity. Finally, we outline recommendations for further research required to develop the identified fly taxa for use in managed pollination services.

## 2. Search Strategy and Selection Criteria

A literature search was conducted using Scopus, Web of Science and Google Scholar databases with the terms “fly pollination”, “fly pollinator”, “Diptera pollination” and “Diptera pollinator” combined with the individual search terms, namely “Australia”, “crop”, “horticulture”, and terms for specific crops of interest, namely “avocado”, “berry”, “berries”, “mango”, “lychee”, “vegetable”, “seed crops”, “carrots” and “brassica”. The crops of interest were selected based on (i) their prominence in Australian horticulture, (ii) their known requirements for insect pollination to improve yield and (iii) their commercial production representing a range of cropping systems and growth environments. Search results were then filtered at three levels (i) title and journal, (ii) abstract, and (iii) full text. From over 1185 papers matching the search criteria, only appropriate articles and relevant references cited therein, were reviewed. Studies were excluded if they were not written in English, or reported dipteran-plant visitations unrelated to food crops and/or if the results only indicated species presence at the field site, rather than specific contact with flowers. Additionally, Google was used to search for relevant mainstream media reports, media releases and species distribution reports across Australia. Candidate fly taxa were then reviewed in the same manner, using family or species name combined with the search terms, “morphology”, “foraging”, “life history” and “distribution” to assess characteristics relevant to pollination potential. A total of 3976 papers were returned on the identified species and reviewed according to the above criteria, with a specific focus on data relevant to the development of each species as a pollination service. 

## 3. Fly Taxa Associated with Horticultural Crops

Flies from 86 families of Diptera have been reported visiting the flowers of more than 1100 different species of plant. Not all of these visitations necessarily contribute to pollination, but the diversity and frequency of associations observed indicates the likely contribution of numerous fly species to global plant pollination [22]. In terms of agricultural food production, many fly species are specifically involved in crop pollination [7] and are known to increase yields [23]. Numerous fly families have been recorded visiting horticultural crops (Table 1), including Calliphoridae (blow flies), Syrphidae (hover flies), Sarcophagidae (flesh flies), Muscidae (house fly and relatives), Rhiniidae (nose flies), Bibionidae (march flies), Anthomyiidae (flower flies), Bombyliidae (bee flies), Stratiomyidae (soldier flies), Tachinidae (bristle flies) and Tabanidae (horse flies). Flies that can be effective pollinators include blow flies (Calliphoridae), hover flies (Syrphidae), flower flies (Anthomyiidae) and, specifically, the house fly, *Musca domestica* Linnaeus 1758 (Muscidae) [24]. Our understanding of the role of flies in pollination remains limited as Diptera are often overlooked in pollination studies and, consequently, their pollination abilities and relevant biological attributes have yet to be exploited [12,13,25,26].

## 4. Target Horticultural Crops

Representative crops from the most prevalent Australian cropping systems were selected for assessment of flies as potential providers of crop pollination services across a range of conditions. Different cropping systems present insect pollinators with distinct environmental conditions, including differences in layout and semi-enclosure design and materials, which are likely to impact foraging behaviours and pollination abilities. Including a selection of representative crops from the different primary cropping systems in our review provides a comprehensive picture of existing fly pollination evidence under common growing conditions in Australia. 

Target crops comprised avocado, mango and lychee trees representing open orchards, berries (blueberries, strawberries and raspberries) and vegetable seed crops (broccoli, Brussels sprout, cabbage, carrots, celery, leek, onion, pak choi and peppers) representing both open and protected (polytunnels, greenhouse and glasshouse) cropping systems. Studies involving these target crops were reviewed for any evidence that flies play a role in pollination in key growing regions, with the view to quantifying pollination effectiveness and assessing the potential of fly taxa for use as commercial pollination agents. 

### 4.1. Avocado (Persea americana)

Avocado flowers contain 5,000–10,000 pollen grains each and, within the flowering period, a mature tree may support over a million flowers. The limiting factor in their successful pollination is synchronizing male and female flower opening times [59]. Consequently, insect pollination is highly valuable as avocado pollen has a long viability following dehiscence from the flower anthers (e.g., ~151 hrs under natural conditions for ‘Simmonds’ cultivar from California). Flower-visiting insects that forage actively both throughout the day and across days are important vectors for pollination as the pollen remains viable while transferred, even though the timing of the opening of male and female flowers in a crop may not overlap [60]. 

Honey bees are usually considered to be the most important pollinators of avocados [61,62,63]. However, avocado flowers are not particularly attractive to honey bees compared to many competitor plant species (sympatric plants with shared flowering synchrony), resulting in sub-optimal fruit set [64]. Wild non-bee pollinators (including blow flies) have been recorded visiting flowers [45,47] and in southern Mexico and Central America where avocados are native, the large, green blow fly, *Chrysomya megacephala* Fabricius 1794 (Calliphoridae), was reported to contribute significantly to pollination [45]. In this study, the blow fly visited more flowers within a given time than other pollinators, occurred in high numbers and was observed with pollen grains on areas of the body that would contact the avocado flower stigma during nectar feeding [45]. In Australia, the cosmopolitan fly, *Ch. megacephala,* has a wide distribution across all states except Victoria and Tasmania and occurs across a range of habitats including bush, farm and urban centres in relatively high abundance throughout the year [65]. A recent review of avocado pollinators in Australia, however, identified a bristle fly (Tachinidae) along with honey bees as being the most efficient insect pollinators in terms of delivering the most pollen to stigmas per minute [32]. 

Less efficient pollinators may still play an important role. Several studies have indicated that hover flies (Syrphidae) and blow flies (Calliphoridae), despite being less efficient at transferring pollen on an individual visit basis, were more effective pollinators overall than bees because of their relative abundance and duration of foraging [36,66,67]. Blow flies were identified as the insect most frequently visiting avocado flowers, as well as being the dominant pollinators of avocado in the Tri-State region of Australia [36,67]. Specific blow flies (Calliphoridae) identified included *Calliphora vicina* Robineau-Desvoidy 1830, *Calliphora stygia* Fabricius 1781, *Calliphora augur* Fabricius 1775 and *Chrysomya rufifacies* Macquart 1842 [32]. As successful pollination is dependent on synchronization of insect activity and floral receptivity, aligned foraging behaviour of many calliphorids with avocado flowering periods strongly indicates the potential for these taxa as managed pollinators. 

Climatic conditions, primarily temperature, greatly impact anther dehiscence, stigma receptivity and fly foraging, suggesting that assessment of insect activity and crop flowering period are also geographically and climatically dependent. Thus, insect species that frequent flowers, are widely distributed and have a broad active thermal range offer the greatest potential as alternative pollinators. Representatives from the fly families, Calliphoridae, Rhiniidae and Syrphidae, are common wild pollinators across multiple crop types and regions of Australia, including avocado and mango [37]. In particular, the rhiniid, *Stomorhina discolour* Fabricius 1794 (Rhiniidae), was as numerous as, or co-dominant with, two bee species across an Australian pollination network spanning regions, years and crop types including avocado, mango and macadamia. These three dipteran families provide shared pollination services across Australian crops and regions and thus offer considerable potential for development as managed pollinators [37].

### 4.2. Mango (Mangifera indica) 

Mango is native to Northeast India and is a major fruit crop grown throughout the tropics. Cross-pollination is generally beneficial but not always essential, and there is variation among the many cultivars. Nevertheless, pollinators are required to move pollen to stigmas, even when self-pollination is viable. Diverse insects visit mango flowers, including many species of bees, wasps, flies, beetles, ants and butterflies [37,43,68,69,70]. Honey bee hives are sometimes employed in mango orchards and honey bees visit mango flowers (e.g., Carvalheiro et al. 2010, Sung et al. 2006), but often prefer other floral resources.

Studies in multiple countries suggest that flies are important in mango pollination. For example, in Taiwan, two flies (*M. domestica* and *Ch. megacephala*), alongside honey bees, were the key pollinators [70]. Meanwhile, a recent study in Malaysia [71] found that the most abundant and effective pollinators of mango were large flies in the genera *Eristalinus* (Syrphidae) and *Chrysomya* (Calliphoridae). Similarly, in Israel, the calliphorids, *Chrysomya albiceps* Wiedemann 1819 and *Lucilia sericata* Meigen 1826, along with the house fly, *M. domestica* (Muscidae), play a significant role in mango pollination, where these fly taxa are as effective as bees [50]. Studies in India and Pakistan also highlight the significance of Calliphoridae in particular [34,69], which had a positive impact on mango fruit quality and quantity [34].

In Australia, there are many *Calliphora* species (Calliphoridae) and *Ch. megacephala*, in particular, was identified as a pollinator of mango in the Northern Territory [43]. Other studies in two regions of Queensland [37] reinforce the general view that Calliphoridae and Syrphidae are significant for mango pollination. However, the single most important pollinator was *S. discolor* (Rhinidae), which was both highly efficient in single-visit pollination and highly abundant [37]. Another rhiniid species, *Stomorhina xanthogaster* Wiedemann 1820, was recorded in earlier surveys in the Northern Territory [43]. The strong evidence for the importance of flies has already led to some attempts to boost or support fly populations on mango orchards. For example, In India, blowflies (*Lucilia* sp., Calliphoridae) and fleshflies (*Sarcophaga* sp. Sarcophagidae) have been reared and released into mango orchards [69]. Fish or mutton pieces have been hung from mango branches in bags to support fly breeding, and we also encountered this “carrion approach” in use by mango growers in some parts of Australia in 2019 [72].

### 4.3. Lychee (Litchi chinensis)

Lychee is a highly cross-pollinated crop and needs insects to assist with pollination [73,74]. Flowering, although variable between growing regions, usually occurs over six weeks between July and October in Australia. Self-pollination by wind can occur, but flowers are typically self-sterile and require insects to transport pollen. Various insects have been reported to visit lychee flowers, including beetles, bugs, moths and flies; however, honey bees are often considered to be the principal pollinator [75]. In Australia, both honey bees and stingless bees (Apidae) are found on lychee blossoms, but preliminary studies suggest that native stingless bees (*Tetragonula* spp.) are too small to be effective pollinators [75]. Honey bee pollination can significantly increase lychee yield, although yields can sometimes be unreliable and rarely approach the capacity of the tree, suggesting that honey bees may be inefficient in this context and/or an abundance of insect pollinators is required [75]. The potential role of dipteran pollinators within lychee crops remains unknown. 

### 4.4. Berries (Blueberries, Strawberries and Raspberries)

Insufficient insect pollinators during flowering is often the cause of poorly formed and misshapen fruit in berry crops. Commercial blueberry (*Vaccinium* spp.) production is virtually entirely dependent on honey bees for pollination [76]. Other recognised blueberry pollinators include mining bees (Andrenidae), bumble bees (Apidae), sweat bees (Halictidae), and mason bees (Megachilidae). Campbell et al. (2017) reported that caging the bumble bee, *Bombus impatiens* Cresson 1863 (Apidae), at high density with highbush blueberry did not increase fruit set or fruit weight compared to open pollination. Similarly, supplemental bees (up to 12.5 colonies/ha) did not increase fruit set in rabbiteye blueberry bushes [76].

There is very little information available regarding fly pollinators of blueberries [33,77,78] or raspberries, either in Australia or elsewhere. Mann (2014) suggested that blow flies (Calliphoridae) may be effective pollinators as they pollinate by sonically vibrating the blossom (i.e., buzz-pollination as seen with bumble bees), which is essential for successful blueberry pollination [42].

Strawberry (*Fragaria* spp.) plants produce aggregate fruit, which allow an assessment of the pollination on each individual berry; incomplete pollination results in malformed fruit. The flower’s exposed nectaries attract a diverse array of flower-visiting insects. Native wasps and bees are involved in strawberry pollination [79], including the solitary mason bee, *Osmia lignaria* Say 1837 (Megachilidae), which was recently shown to increase strawberry yield when released on small farms [80]. In greenhouse environments, stingless bees have been shown to be effective pollinators of strawberries [81,82]. The hover flies (Syrphidae), *Episyrphus balteatus* De Geer 1776 and *Eupeodes latifasciatus* Macquart 1829, have also been demonstrated to be effective pollinators of commercial strawberry crops with flower visitations by these species improving fruit yield by >70% and increasing the proportion of marketable fruit [83]. Single visits by large hover flies have been reported elsewhere to be as important to strawberry pollination as have visits by honey bees and small, solitary bees [10].

Similarly, insect pollination is vital for optimization of raspberry (*Rubus* spp.) production with additional benefits of insect visitation contributing to a reduction in mould and fungal outbreaks. Raspberry flowers contain more than 100 pistils and each must be pollinated in order to create a mature seed and drupelet surrounding the seed. A raspberry fruit is composed of 75–85 drupelets, with any unpollinated drupelets compromising the overall integrity of the fruit; often many drupelets at the tip of the fruit are not pollinated and remain immature and small [84]. Aside from impacting pollination, insufficient insect visitation can result in excess nectar accumulating on flowers, which can cause problems such as sooty mould and other fungi growing in the sugary exudate [85]. Little information is available regarding insect pollinators in raspberry crops. Andrikopoulos and Cane (2018) compared the pollination efficacy of five bee species on raspberry, but there are currently no reports of fly visitation available.

### 4.5. Vegetable Seed Crops

Vegetable crops are grown not only for the purpose of food production, but also to supply seed for propagation. Many of these seed production crops are reliant upon insect pollination. The main insect-pollinated vegetable seed crops grown in Australia are carrots and other Apiaceae; vegetable brassicas (Cruciferae), including cabbage, cauliflower, broccoli, kohlrabi, brussel sprouts (all *Brassica oleracea*) and Asian greens (*B. rapa* and *B. napa*); and Alliaceae including bulb onions (*Allium cepa*), bunching onions (*A. fistulosum*) and leeks (*A. porrum*). These crops typically feature small, unspecialised white, yellow or green flowers clustered in umbels (Alliaceae and Apiaceae) or racemes (Cruciferae), and are predominantly pollinated by honey bees, both managed and wild. However, a diverse assemblage of other insects including flies, native bees, wasps and beetles also contribute to pollination in open field crops [14,29,86], where most vegetable seed is grown. In other settings such as cages, glasshouses or polytunnels, which are commonly utilised for the production of breeding lines and small, high-value seed lots, flies can be used as an alternative or supplement to honey bee pollination [38].

Much of the vegetable seed grown in Australia is hybrid seed, which is produced by pollinating a seed bearing (female) parent line with pollen from a pollen donor (male or pollinator) parent line. Male and female hybrid seed parent lines are typically grown in alternating 1 m to 10 m wide strips within the crop to facilitate pollen transfer and separation of the parent lines during harvest. To ensure hybridisation, the female line is either female seed-bearing (does not produce pollen) or features strong self-incompatibility (produces pollen but is largely incapable of self-fertilisation) [87].

The prevalence of hybrid seed production in Australia is important because inadequate pollination can be a yield-limiting constraint in many hybrid vegetable seed crops, including carrot [38,88], *B. oleracea* [87] and onion [89]. Underlying factors restricting pollination in these crops include the need for insects to move from a pollinator to a male sterile plant to effect pollination, and differences between hybrid parent lines that elicit discriminatory foraging behaviour in honey bees. This often results in one parent line being visited to the exclusion of the other because of variation in flower colour and morphology, nectar production rates and/or nectar quality [88,90]. In addition, honey bees sometimes find carrot and onion seed crops less attractive than other nearby forage sources, and thus leave the target crop to forage elsewhere [89]. In protected crops, enclosure coverings can also compromise the ability of honey bees to navigate and forage normally [91,92]. Given the substantial reliance on managed honey bees to pollinate vegetable seed crops, and the challenges that this can present for achieving reliable yields in some situations, efforts to develop alternative pollinators for use in this context are warranted.

After bees, flies are one of the most common pollinators to visit vegetable seed crops [14,29,54,86,93], and thus, they are an obvious target for research in this area. To date, attention has focussed largely on the use of flies for mass pollination in covered systems, including individually bagged plants, small cages, polythene tunnels and glasshouses. In caged plots, onion seed yields from plants pollinated by the blow flies *Calliphora vomitoria* Linneaus 1758*, Lucilia caesar* Linneaus 1758 and *L. sericata* (Calliphoridae) were comparable to, or better than, yields from hand-pollinated or honey bee-pollinated plants [94,95]. Similarly, the blow fly, *C. vicina,* and house fly, *M. domestica*, have been shown to be effective pollinators of caged leek [39] and carrot [40,96] plants. In addition, covered plots of self-incompatible *B. oleracea* seed parent lines pollinated by *C. vicina* demonstrated improved hybridisation rates and reductions in undesirable sib (self-pollinated) seed production compared to those pollinated by honey bees, because the blow flies exhibited more random foraging pattern across parent lines whereas honey bees selectively foraged individual lines [31].

Notably, while honey bees were better pollinators of carrots than house flies, *M. domestica*, a combination of both insects produced more carrot seed per cage than either honey bees or house flies alone [96]. This, combined with Howlett’s (2012) observation that *C. vicina* was most abundant in carrot fields when temperatures were lower than optimal for honey bees, highlights the potential advantages that the use of multiple, complimentary pollinators could offer for vegetable seed crop pollination. 

Flies are also successfully employed as pollinators for larger-scale bulking of vegetable seed in tunnel houses. Schreurs and Sons in Victoria, Australia, use *L. sericata* (Calliphoridae) to produce celery seed and Tasmanian farmer Alan Wilson employs blow flies including *Lucilla* and *Calliphora* spp. reared on carrion, in combination with honey bees, to produce hybrid cauliflower seed [97]. 

To date, comparatively few studies have examined the potential for flies to be used as managed pollinators in open field crops. Animal carrion is sometimes introduced into vegetable seed crops in the hope that the flies that hatch from it, along with those that are attracted to the field by its presence, will contribute to pollination. However, information regarding the timing of carrion introduction and any impact on pollinator species composition, pollination and seed yields in open field settings is lacking. Some evidence exists related to the non-carrion breeding drone fly, *Eristalis tenax* Linnaeus 1758 (Syrphidae), which has been identified as an important pollinator in open crops of hybrid carrot, pak choi (*Brassica rapa* subsp. *chinesis* (L.) Hanelt) and onion seed in Tasmania and New Zealand [14,38,41]. In these crops, it displays pollination efficiency traits similar to honey bees in terms of pollen loads carried [38] and stigmatic pollen deposition [41], but it also crosses more readily between hybrid parent lines than honey bees [51]. However, while *E. tenax* has been demonstrated to pollinate field crops and displays some life cycle, physical and behavioural attributes desirable for mass rearing on cheap and easily obtained substrates, specific information relating to its management as a field crop pollinator is currently lacking [58].

An important consideration surrounding the potential use of flies as managed pollinators in open vegetable seed crops is the issue of insect pest control. Although an essential component for producing high quality seed, pesticide applications are known to have negative impacts on populations of wild pollinators including flies [29]. Strategies to retain fly pollinators in open field crops are needed to avoid unintended negative consequences of mass fly releases on surrounding land users and the environment. Additionally, the development of flies as alternative pollinators of vegetable seed crops will also require innovation of pest control practices and/or pollinator species selection and release strategies to mitigate potential losses incurred through pesticide application.

## 5. Identified Dipteran Pollinators

A review of the pollination literature relating to avocado, berry, mango, lychee and vegetable seed crops revealed that representatives of three key families, the Calliphoridae, Rhiniidae and Syrphidae, are the most commonly reported fly taxa to visit crop flowers and/or play a role in crop pollination (Table 1). Similarly, a quantitative evaluation of 39 studies of crop pollinators worldwide by Rader et al. (2020) identified syrphids and calliphorids as the two most significant taxa of non-bee pollinators of pollinator-dependent crops globally. However, beyond reports of horticultural crop visitations by representatives of these families, there is little specific information regarding the foraging ecology and pollination abilities of the relevant species. It is possible, nevertheless, to relate existing knowledge of these families to the morphology, physiology, behaviour and life history traits associated with crop pollination success, to aid in the identification of fly taxa likely to be suitable for managed pollination applications within Australia. Here, we provide a conceptualised framework of considerations relating to species-specific morphology, behaviour and life history traits as a guide for this process both within Australia and regardless of locality (Figure 1).

The efficiency and effectiveness of any given pollinator is measured by the ability of that insect to move and deposit pollen between flowers. Pollination effectiveness is measured as the seed set following a single flower visit by a pollinator and can be further quantified as single visit pollen deposition (SVD), defined as the number of pollen grains deposited on a virgin stigma in a single visit [98,99]. Pollinator species vary in their efficiency and effectiveness due to morphological, physiological and behavioural differences. Morphological traits, such as mouthpart-characteristics, body mass, body length and the amount and location of pilosity (hairiness) on pollen-contacting body parts are likely to be good predictors of pollinator efficiency and effectiveness [22]. Stavert et al. (2016) developed a method to measure insect hairiness as a predictor for SVD. Facial hairiness was a strong predictor, explaining more than 90% of the variation in SVD measures on pak choi and kiwi fruit, with the hairiness of ventral and dorsal thoracic regions also being good predictors depending on the flower type. Hairiness must be assessed in the context of the flower structure and the way the insect interacts with the flower, concentrating on the body regions identified as most frequently contacting the flower’s reproductive structures [20]. Sexual dimorphism in both hairiness and/or the way the individual interacts with the flower must also be considered. Such morphological traits must align with the reproductive structural morphology of the target crop, including flower shape and nectar and pollen location [20,100].

Similarly, the foraging ecology of the pollinator species must align with the floral ecology of the specific crop (timing of stigma receptivity, anther dehiscence and pollen viability), which can be complex (e.g., avocado). The phenological synchrony of crop-pollinator interactions (flowering times of different crops and the activity of insect pollinators) is driven by abiotic and biotic factors, including temperature, humidity and insect physiological status [101,102]. Sex, reproductive status and/or life stage can influence foraging ecology in response to fluctuating nutritional requirements and should be considered when assessing the synchrony of crop and potential pollinating taxa [98,103,104]. Suitable pollinating taxa must have a distribution that coincides with target crop production areas and demonstrate seasonal synchrony between flowering time and insect visitation/activity [105]. Fly taxa with foraging ecology distinct from that of bees can potentially improve pollination and yield through complementary pollination [28].

Fly foraging behaviour may be comparatively less likely to cause pollination compared with honeybees. However, both taxa actively search for and consume pollen/nectar and also carry loose pollen on their body. Some solitary bees may have a lot of loose body pollen, but honeybees and stingless bees carry most food pollen in sticky pollen balls that do not contribute to pollination. Flies may prove better able to contend with variable conditions (e.g., remain active over a wider temperature range) impacting their activity and, thus, potentially complement bee pollination services. Ellis et al. (2017) reported that while honey bees were the most abundant and important pollinator of strawberries, numbers declined as the long flowering season progressed and flies (Syrphidae and other unidentified fly taxa) contributed more significantly to pollination in the later stages of the flowering season. This was also the case during periods of poor weather (increasing wind, rain and low temperatures), when flies remained abundant while bee numbers declined [105]. Such reports suggest that fly taxa foraging behaviour can be aligned with flowering seasons for specific crops and can likely augment existing pollination services. Investigation of the daily timing and duration of pollinator foraging activity is an important consideration in identifying suitable managed pollinators for a target crop.

Aspects of commercial crop production such as undercover cropping, including polytunnels and glasshouses, are also relevant to the foraging behaviour of fly taxa. At high altitudes and in subarctic climates, flies are acknowledged to play a greater role in pollination than bees, being of comparatively higher abundance and typically demonstrating greater thermal tolerance and physiological plasticity [12,28,106]. The ability of fly taxa to remain active across a wide thermal range, in shaded, overcast and enclosed/covered conditions, has the potential to improve pollination where protected cropping is in place and bee activity may be compromised [28,105,107]. The responses of different insect taxa to the variable conditions created by protective cropping require further investigation, particularly differences in temperature and humidity. Existing studies of fly activity indicate that their foraging behaviour may be more adaptable to a wider range of managed cropping conditions than with bees, providing a potential means of optimising pollination services in environments that are less conducive to bee activity.

The foraging behaviour of insects, and, by extension, the frequency of flower visitation, is driven by the metabolic needs and physiological state of the insect [108,109]. A variety of cues and rewards are used to attract insect visitors to assist in plant reproductive success [110,111]. Colour and odour cues are primarily used by insect taxa to identify required nutritional resources such as sugar (nectar) and protein (pollen) [112]. Colour vision in flies is poorly characterised but appears highly variable across species, necessitating the species-specific assessment of colour attraction preference and crop type. Although the effects of odour cues on flies are better understood, the majority of studies have examined these in the context of locating breeding resources rather than the acquisition of nutritional resources such as sugar, protein and carbohydrates [103,113,114].

## 6. Calliphoridae (Blow Flies)

Calliphorids have a worldwide distribution, with more than 1,000 species and approximately 150 genera described [115] and, although limited empirical evidence exists, they are considered to be important pollinators globally [34]. Alongside both managed and wild bees, blow flies are likely the main crop pollinating insect [23,116]. Calliphorids visit a broad range of flowering plants, including many crops, and may pollinate a wide variety of flowers, having been observed in the wild with large amounts of pollen on their bodies [19]. Some species, being nectar feeders, regularly visit certain plants for sugars and must inadvertently cause pollination; however, it is not known how their pollination abilities compare to those of bees. Despite being one of the taxonomic groups most commonly associated with crops, studies assessing the contribution of calliphorids to pollination outcomes have been few and, where reported, the specific species involved in pollination activities are often not identified beyond genus (Table 1).

### 6.1. Morphology

Significantly, blow flies possess an assortment of morphological traits that are typically associated with successful pollination abilities. The majority of blow flies have an extending proboscis with sponging or lapping mouth parts that necessitate broad contact of the fly’s head and upper body with the inside of the flower [117]. Many of the flowers produced by crops of interest are accessible to blow flies; avocado nectar, for instance, is readily accessed by the lapping mouth parts of *Ch. megacephala* [45].

Calliphoridae also have numerous body setae (hairs), often of diagnostic importance, that vary in density and location with species type [118]. The pilosity of pollen-contacting body parts is likely a good indicator of pollen load (the amount of free pollen adhering to the insect’s body) and potentially facilitates pollen deposition when considered in the context of insect-flower interaction [20]. In calliphorids, extensive setae are located on the head and thorax [118], providing the potential for pollen trap and transfer between flowers.

The relatively robust body size and shape of calliphorids, compared to other slender and more delicate fly taxa, may also enhance pollination by potentially increasing the surface area of the body coming into contact with pollen and stigmatic surfaces during feeding. In addition, their greater weight, similar to honey bees, may trigger access to concealed pollen or nectar in some plant species [98,119]. Given that body size can be a predictor of SVD, variation in this attribute can be important at both the intra- and inter-specific species level. Notably, calliphorid species, such as *C. vicina* and *Ch. megacephala*, demonstrate a high level of phenotypic plasticity in body size, along with various other life history traits [120,121].

Nutrition during larval development is the primary determinant of the size of various morphological traits in holometabolous insects, including adult body size, although numerous environmental factors can play a role [122,123]. Where food resources are limited and/or overcrowding restricts access to them, individual blow fly larvae may satisfy their minimal nutritional requirements to complete development but will develop into relatively small adults [21]. Alternatively, where larval food intake exceeds the minimum nutritional requirements, the arising adult is comparatively larger, which confers several reproductive advantages. However, the requisite extended period of feeding also has the disadvantage of prolonging vulnerability to predation and parasitism during the larval stage [121,124,125]. Aside from total food intake, the ration of fat, protein and carbohydrate in the larval food resource also influences adult body size, with blow flies demonstrating a species-specific response to different diets. In one example, high fat diets were linked to comparatively smaller adult *Ch. maegacephala* [120]. Importantly, adult blow fly body size, which is subject to various abiotic and biotic factors under natural conditions, can easily be manipulated within a controlled rearing facility by altering dietary conditions during larval rearing [21,124]. The suitability of calliphorid taxa, such as *C. vicina* and *Ch. megacephala* for use as pollinating taxa can thus be potentially tailored to optimise pollination efficiency by manipulating adult body size to match the flower shape/size of specific target crops. 

Manipulation of blow fly body size may also optimise other attributes that enhance pollination efficiency. For instance, body size appears to influence the foraging activity of *Ch. megacephala,* with larger flies active earlier in the morning and at lower light levels, than smaller individuals [123]. Flower-foraging insects have a large number of ommatidia, which are thought to confer heightened visual acuity, movement awareness and depth perception [126,127]. Correlation between insect body size and the number of ommatidia occurs in both bees (Apoidea) and flies (Calliphoridae). For *Ch. megacephala,* eye size scales proportionately to body size and investigations by Smith et al. (2015) attributed differences in the foraging behaviour between flies of different sizes to the positive relationship between total eye area and body size, which allows for increased light capture and activity at lower light levels with increasing eye area [123]. The positive relationship between metabolic rate and body size is also likely a contributing factor to variation in foraging activity patterns throughout the day [128]. Although comprehensive research on the visual efficiency of blow flies is lacking, such evidence suggests manipulation of fly body size during rearing and selection of larger blow flies for pollination services would be desirable, due to heightened visual efficiency during flower foraging and potentially extended flight activity during periods of lower light levels [123].

### 6.2. Distribution and Foraging Behaviour

Within Australia, there is a diverse range of calliphorids, with many taxa demonstrating a broad distribution and high abundance levels throughout the year. Selection of pollinating agents for managed release must consider any potential for negative interaction with existing biodiversity, particularly other pollinators. Regionally-specific fly taxa, already occurring in both natural and managed Australian environments, are clearly preferable candidates for managed pollinators. Several key blow flies occur in areas of target crop production in Australia and demonstrate seasonal synchrony of flowering time and insect visitation/activity (Figure 2; Table 2). Horticulture in Australia occurs primarily within the coastal regions of Queensland, central and eastern New South Wales, southern Victoria, northern and south eastern Tasmania, southeast South Australia, southern Western Australia and the Northern Territory [129]. Thus, there is considerable synchrony between potential fly pollinators and key crop distribution (Figure 2). While some of these species are native to Australia, many are found globally such as the ubiquitous, *Ch. megacephala*, which has been reported to pollinate avocado and mango and occurs widely across Australia [130]. 

One of the most efficient pollinators of *Medicago citrina* (a wild relative of lucerne) is the blow fly *C. vicina* [177], which has been identified as a potential managed pollinator since it occurs globally and is easy to rear [40]. The species was recorded as a pollinator of *Hebe* spp. (grass) and at least 18 species of exotic plants nearly a century ago [178], supporting its likely value as a general pollinator of numerous flowering species including crops. Within Australia, *C. vicina* is distributed throughout the south-eastern states including Tasmania. Recently, the species has also been identified in traps in the south-west of Western Australia (D. Cook pers. comm. 20 Nov), indicating a wide distribution across Australian horticultural regions (Figure 2). Endemic species in the same genus may offer similar potential, including *Calliphora dubia* Macquart 1855, *Calliphora albifrontalis* Malloch 1932 and *Calliphora varifrons* Malloch 1932. Representatives of the *Chrysomya* genus, including *Ch. rufifacies*, *Chrysomya saffranea* Robineau Desvoidy 1830 and *Chrysomya varipes* Macquart 1851, occupy an even broader distribution and typically have a higher active thermal range applicable to Northern Australia [157,179]. Given the ubiquitous overall distribution of blow fly species, Australia is well placed to identify commercially viable species for release and pollination services (Figure 2).

Along with the flowering times of different crops, the activity of insect pollinators is driven by temperature, radiation intensity and photoperiod. While the presence and abundance of a species is strongly governed by spatial and temporal environmental fluctuations, the diversity within the calliphorid taxa allows for the alignment of species seasonality (and thus activity upon managed release) with specific crop flowering times [28]. Synchrony can be achieved by selecting species that are already present in areas of horticultural interest and have been identified as local pollinators. As managed pollinators, the stocking density and time of release to target crops could then be optimised to augment pollination.

Blow flies demonstrate high plasticity in their thermal tolerances and, by extension, their active thermal range is typically broad, facilitating flight activity under a wider range of conditions than bees [28,40,105,180]. Honey bees (*A. mellifera*) have a minimum temperature threshold of 8.7–11.2 °C for foraging activity, whereas the active thermal range of *C. vicina* is estimated to fall between 4–25 °C [134,181,182]. Rader at al. (2013) reported differing patterns of foraging activity between flies and bees throughout the day. Flies were active at times when bees were absent or present only in low numbers, such as during overcast conditions and/or early and late in the day. Consequently, where environmental conditions fluctuate, pollination outcomes may benefit from diversification of foraging activity across managed pollinators [15]. 

Bimodal curves describing daily flight activity with peaks in both the morning and afternoon were reported for many species of blowflies based on trap catches by Norris (1966). Notably, *Calliphora* blow flies were active for longer periods of the day than *Chrysomya* or *Lucilia* species [183]. One species, *C. stygia*, was active all year, and during spring and summer, daily activity exceeded 15 hours. More recently, Lutz et al. (2019) identified six environmental factors as important determinants of blowfly flight activity, including ambient temperature, solar radiation, humidity, rainfall, barometric pressure and wind speed. Responses to abiotic factors were species-specific and, given the importance of temperature, largely determined by the lower temperature threshold of a species [163]. Individual species differ in their active thermal range; however, within species, geographic variation in seasonal thresholds of temperature and illuminance is also evident [183]. In this context, investigation of the thermal requirements of calliphorid species, where unknown, would prove valuable in assessing the suitability of a species as a pollination agent and optimising release practices for a given geographic cropping region.

Foraging bees rely on the sun as a primary reference for navigation and communication, and thus, cloudy conditions and protective cropping systems can negatively impact on foraging activity [91,92]. In contrast, blow flies show potential as pollinators for the greenhouse industry, as they may be less sensitive to heat, humidity, solar radiation and UV distortion than honey bees. Blow flies exhibit a bimodal flight curve of daily activity, influenced primarily by ambient temperature, solar radiation (light intensity) and humidity [183]. The environmental conditions under which blowflies are active, while both species-specific and adaptive, can be considerably broader than those for honey bees. The broader active thermal range of specific blow fly species may provide a beneficial supplemental pollination service across cropping systems, particularly where crops are enclosed or semi-enclosed.

The life history traits of calliphorids thus require species-specific assessment against flowering parameters of target crops to optimise pollination outcomes. The foraging requirements of calliphorids vary with physiological status but, importantly, sugars (carbohydrates) and protein are often obtained from flower visitation [15,28]. Notably, for calliphorids such as *L. sericata,* pollen can be a protein source for egg development, and is likely to play a major role in their foraging ecology. The reported frequency of flower visitations by calliphorids is probably linked to the reward incentive offered by both pollen (protein) and nectar (sugar/carbohydrates for energy demands), indicating likely interaction with, and transfer of, pollen. Calliphorid taxa thus not only possess favourable morphological attributes, in terms of size and hairiness, but also demonstrate foraging behaviour that indicates high potential as pollinators in horticultural settings [112].

Pollination efficiency and effectiveness can vary depending on the sex and/or physiological status of the individual insect [98,103,104]. Flower visitation within the identified Calliphoridae (Table 2), for example, can be to search for either pollen and/or nectar, depending upon the metabolic needs of the individual. Carbohydrates are required to fuel flight and metabolism, while protein is essential for reproductive function and oocyte development [112]. Following eclosion, females do not mate immediately, as they require protein for both sexual receptivity and egg development [139]. Females may thus preferentially feed on pollen, whereas males may preferentially feed on nectar to power flight muscles, allowing them to disperse and thus maximise the likelihood of mate location and reproductive success [104,112]. Conversely, after egg maturation, females may preferentially feed on nectar as increased sugar inputs are reported to improve female lifespan and thus maximise female reproductive output [138]. Males of many calliphorid species, including *C. stygia*, defend ‘mating stations’ at nectar and honeydew sources that attract females [18,144]. Males perch on surrounding vegetation and fly out to intercept potential mates [18,184]. There is evidence that males are more common at carbohydrate sources than proteinaceous ones [184,185], and this may be because the chance of an encounter with a receptive female is higher. Once mating has occurred and females are gravid, their foraging requirements change and oviposition resources are sought [103]. Therefore, establishing the optimal sex and/or physiological status of the calliphorid candidate intended for managed release offers a means of further enhancing pollination efficiency. This may have practical applications for rearing and managed release of the blow fly, *Ch. rufifacies*, where sex is determined by monogenic reproduction and females have offspring of one sex only (all males or all females) [160,186].

Additionally, while the majority of calliphorid species are oviparous (eggs are oviposited), others display ovoviviparity (eggs are hatched within the body and larvae are larviposited) [139,148,150]. Reproductive mode is likely to influence the energetics and nutritional requirements of potential pollinating species, because larvipositing species may spend more time foraging for resources than ovipositors due in part to the comparatively longer period of gestation [187]. For this reason, larvipositing species such as *C. dubia* may prove to have greater pollination abilities than ovipositors such as *C. albifrontalis.* Reproductive mode should therefore be considered when assessing the suitability of potential alterative pollinators. 

As visual and semiochemical cues are used to detect pollen or nectar sources, aspects of floral colour and odour will influence blow fly foraging activity. Brodie et al. (2015), investigating colour and odour preference in *L. sericata,* reported an enhanced attraction to yellow over other colours in young unmated flies when in flower foraging mode. The inclusion of floral odour from Oxeye daisies (*Leucanthemum vulgare* Lam.) also enhanced attraction but fly response was dependent on the presence of a yellow colour cue. The impact of floral colour and odour on foraging behaviour in calliphorids requires further investigation to best match potential pollinators to target crops.

### 6.3. Rearing and Release

While the life history traits of calliphorids are species-specific, taxa are typically easy to rear in colony and feed on commonly available rearing media without any need for parallel rearing of a secondary insect food source [149]. Rearing protocols have already been developed for a number of species [133,188,189], demonstrating the ability for most to be mass reared for commercial applications due to their high fecundity, rapid development (which can be optimised by environmental control) and minimal husbandry costs (Table 2).

Importantly, the relationship between temperature and blow fly development can be manipulated to facilitate the synchronisation of pupation and emergence with transport and application timing [149,188]. The life cycle of the blow fly includes egg, larval and pupal stages with pupation having the longest duration of all the phases. The occurrence of a pupal (static) stage facilitates transport and release practices, providing a convenient form for sending the required stocking numbers to commercial properties. Pupae can then be distributed on site, by placement within suitable emergence houses, where daytime temperature will expedite emergence of adults. With refined processes, the timing of emergence can be linked to transport and release requirements within a specific crop. A further advantage is the ability, under environmentally controlled conditions, to induce diapause during the pupal stage in many blow fly species. The application of low temperatures can be used to induce diapause and cease development until such time as temperatures increase again. Manipulation of temperature can thus be used to optimise development according to rearing and transport requirements, effectively increasing the ‘shelf life’ of pupal stock for distribution purposes [149].

Data detailing blow fly dispersal following emergence, specifically in regard to foraging activity and pollination, are currently lacking. Calliphoridae have good flight ability and can cover long distances relatively quickly: the dispersal of *Ch. rufifacies* has been measured between 1.3–6.4 km/day [144,190], while for *Calliphora nigribarbis* Vollenhoven 1863, it was as high as 1.7 km/day [191]. Although blow flies are capable of dispersing widely, flight activity is driven by nutritional requirements. If the required floral resources are present at the emergence location, it is anticipated that individuals will remain in the area, dispersing only if additional resource needs arise. However, development of calliphorid pollinating agents as a commercial service requires more comprehensive understanding of the species-specific factors influencing dispersal on emergence, to ensure optimal flower visitation soon after release. 

The reported lifespan of the identified candidate calliphorids varies between 24 and 67 days according to the species and whether measurements were based on field or laboratory individuals (Table 2). The adult fly life span is dependent on temperature and nutritional intake but, under field conditions, is likely to average approximately 4 weeks. This is relevant when determining stocking rates, the timing of release and the required number of releases throughout the pollination period to optimise species and crop synchrony.

Species being considered for mass rearing and release that are already widely distributed across intended release locations must also be assessed in terms of pest status. Species such as *Lucilia cuprina* Wiedemann 1830 are associated with high levels of animal myiasis and are therefore a poor choice as commercial pollinators, regardless of their pollination ability [192]. Other blow fly species that are considered a minimal nuisance to livestock and humans are still viable options, as the potential exists to sterilise them before release through the established sterile insect technique applied in the control of Mediterranean fruit fly, *Ceratitis capitate* Wiedemann 1824 (Tephritidae) [193]. Non-gravid females would be less likely to disperse in search of oviposition resources and concerns around fly breeding/abundance beyond the pollination period could therefore be mitigated.

## 7. Rhiniidae (Nose Flies)

The Rhiniidae family, previously a sub-family of the Calliphoridae, comprises around 50 genera, of which four, *Stomorhina*, *Chlovorhinia, Metallea* and *Rhinia*, are native to Australia. Recent reports of *Stomorhina* feeding on flowers and demonstrating effective pollen transfer in a number of crops suggest that some species from this genus may be important pollinators of horticultural crops in Australia [37,71,194]. At present, beyond taxonomic descriptions, basic biological and behavioural data relating to Australian taxa are limited but similarities with the Calliphoridae are likely, given the previous taxonomic status of the Rhiniidae as a subfamily within the Calliphoridae.

### 7.1. Morphology

Most rhiniids are small to medium sized flies in comparison with the larger calliphorid blow flies. Often referred to as the wasp-mimicking fly due to abdominal banding, *S. discolour* ranges in body length from 4.5–6.5 mm, while *S. xanthogaster* ranges from 9.5–11 mm in body length. The group are considered hairy, with numerous setae reported along the thorax and other body regions [170].

### 7.2. Distribution and Foraging Behaviour

Distribution and morphological data pertaining to the Rhiniidae are available through published surveys, description of species and related taxonomic revisions, but there is a paucity of specific life history and behavioural information particularly in regard to Australian taxa. Representatives of the genus *Stomorhina* occur across Australia, with many reports only to genus level (Figure 3). Of the two *Stomorhina* species associated with target crops, *S. discolour* has an Austo-Oriental distribution and is established along the east coast of Australia [170], while *S. xanthogaster* has been reported in QLD and NSW (Figure 3). Their foraging behaviour is largely unknown, although in Northern Australian mango crops, *S. xanthogaster* feeds on pollen, contacting the reproductive structures of the flower in the process [43]. Recent identification of the importance of *S. discolour* as an unmanaged pollinator of avocado, mango and macadamia crops in Australia highlight the need to further develop basic biological data for the Rhiniidae taxa [37].

### 7.3. Rearing and Release

Data labels detailing collection information of Australian museum specimens indicate that some rhiniids may breed in termite mounds [170]. Worldwide, representatives of *Stomorhina* breed in locust egg-pods and are associated with the nests of Isoptera (termites) and Hymenoptera (ants), where they are probably ectoparasites [170,195]. *S. discolour* larvae, for example, are reported to feed on the broods of both ants and termites to complete their development [170,195]. Beyond very limited reports, usually associated with species descriptions, there is an absence of published rearing protocols or studies documenting specific life history data of the taxa. A greater understanding of the life cycle of different species is needed to assess their specific potential as managed pollinators. Australian rhiniids are not considered pests but any concerns of increased pest status created by mass rearing and release could potentially be mitigated with sterile insect technique practices. Further investigation of the pollinating potential of the taxa is therefore warranted. 

Breeding of most rhiniids may be host dependent, requiring a secondary invertebrate species to act as larval food source. This would necessitate rearing of both the fly and its host, increasing the cost and complexity of mass rearing compared with the previously identified calliphorids. It is therefore necessary to establish basic life history data and foraging behaviour for key rhiniid species to determine the potential benefit of these flies for managed pollination.

## 8. Syrphidae (Hover Flies)

Syrphidae (hover flies and flower flies) is one of the largest dipteran families with over 6000 species described globally. Syrphids obtain most, if not all, of their resources as adults from flowers with visitation for both nectar and pollen [173]. A diverse range of life histories are represented in the Syrphidae family, with the larvae of many species having an added benefit to horticultural industries as predators of pest insects such as aphids. Thus, some syrphids are considered important pollinators as adults and major biocontrol agents as larvae [13,26]. Drone flies, which are members of the genus *Eristalis* within the Syrphidae, mimic bees and have shown potential for pollination at a commercial scale, particularly in orchards, seed crops and greenhouses [4,56,57]. Whilst generally acknowledged as the second most important pollinator group after bees, there is a paucity of data detailing species-specific biology and the pollination abilities of syrphids [26,173]. As with many diverse fly taxa, identifying specimens to species level is difficult and requires specialist knowledge, which has limited the majority of reports on syrphid flower visitations and pollination activity to family or genus level.

### 8.1. Morphology

Considered generalist foragers, syrphid pollination abilities are likely limited by morphological constraints related to flower size and shape [173]. Hover fly mouthparts are typically short and unspecialised compared to the long feeding apparatus of bees, suggesting a preference for flowers with more open access to floral resources (Gilbert 1981). However, Campbell et al. (2012) reported that while species such as *Ep. balteatus* demonstrated a preference for flowers with a short corolla (<3 mm in depth), other species such as *Eupeodes corolla* Fabricius 1794 (Syrphidae) were more generalist in their flower preference, attending both long and short corolla flowers. There is some evidence of variation in the level of specialisation and flower preference of different syrphid sub-families and species [173]. The relatively short proboscis of species such as *Ep. balteatus* (1.80–2.90 mm) suggests application of syrphid species to managed pollination must ensure the morphology of the insect is aligned with the flower structure of the target crop to be effective [196].

Compared to the majority of calliphorids, hover flies have small to medium body sizes. Body size is considered an important factor influencing the diet of syrphids, with head width often correlated to proboscis length and indicating a taxon’s ability to access flower resources. Gilbert (1981) reported that larger syrphid taxa were more frequent nectar feeders than smaller taxa, which favoured pollen feeding on flower visitation. Potentially, larger taxa sourced energy from nectar and protein from pollen as a consequence of higher metabolic needs than smaller species. With sugar also present in pollen, smaller taxa, with comparatively lower energy requirements, can potentially acquire sufficient energy and protein for reproduction from pollen feeding alone. The occurrence of different foraging strategies among syrphids suggests that larger species may be better pollination agents for horticultural crops. Based on head width, taxa such as *E. tenax* (5.43 ± 0.22 mm) and *Ep. balteatus* (3.07 ± 0.07 mm) are larger species than other possible pollinating syrphids, such as *Syritta pipiens* Linnaeus 1758 (2.12 ± 1.7 mm) or *Melanostoma scalare* Fabricius 1794 (2.11 ± 0.12 mm) [196]. 

Body size also influences foraging behaviour as smaller syrphid taxa prefer large flower inflorescences. This preference may help them obtain their resource requirements from the one location, optimising individual energy use involved in foraging activity [173]. As a larger syrphid species, *E. tenax* is of considerable interest as a pollination agent [58,173].

Many syrphids are covered in fine yellow hairs, which contribute to thermal reflectance properties of the insect cuticle and aid in thermal balance [22]. The hairiness of the insect is likely indicative of its pollination potential, although comparative, empirical measures of hairiness between taxa are currently lacking. In the relatively hairy syrphid, *E. tenax*, pollen particles are trapped within the body hairs before being transferred by combing to pollen-retaining bristles and eventual ingestion [197]. While pollen is consumed by syrphids, the process of collection and also direct feeding behaviour from the anthers inevitably results in pollen transfer from one flower to another through contact with the bristles and surface of the insect during collection and feeding [197].

### 8.2. Distribution and Foraging Behaviour

The Australian Syrphidae taxa are relatively species-poor and poorly studied in comparison to Europe and other regions [198]. Revision of the various genera has frequently occurred with multiple names reported in the literature for Australian taxa being preoccupied and subsequently consigned to synonymy. In particular, the taxa of the northern and western regions of Australia are not well known and more endemic species may be described [199].

Of the syrphid taxa reported visiting flowers, the introduced drone fly, *E. tenax*, has a wide global distribution and high abundance throughout the year [14,58]. Two native Australian drone flies, *Eristalinus punctulatus* Macquart 1847 and *Eristalinus hervebazini* Klocker 1924, are thought to have similar breeding habits and, potentially similar pollination abilities to *E*. *tenax*. Other flower-frequenting syrphids include *Simosyrphus grandicornis* Macquart 1842 and *Melangyna viridiceps* Macquart 1847, which occur throughout Australia and are two of the most commonly observed Australian syrphids (Figure 4). Meanwhile, Finch and Cook (2020) have recently pointed out that the seasonal and geographic patterns of occurrence records for *E. tenax* and *M. viridiceps* probably indicate long-distance migration between different parts of southern Australia. *E. tenax* is known to migrate in Europe and migrations of other hoverfly species have also been recorded, mainly in Europe and North America to date [200].

While reported in various species lists across Australia and associated with flowering crops, existing biological and behavioural data relating to Australian syrphid species are limited or broadly reported to genera, with the notable exception of *E. tenax*. The drone fly, *E. tenax* feeds primarily on nectar, frequently returning to feed on previously visited flowers under caged conditions where space or resources are restricted [57]. Jarlan et al (1997b) viewed the unpredictable dispersal of drone flies (Syrphidae) following release into the field as a possible limiting factor for their application as managed pollinators. In contrast, under caged or greenhouse conditions, *Eristalis spp.* are efficient pollinators of crops such as onion and sweet pepper, where they have been reported to carry more pollen grains than nectar-collecting honey bees of equivalent size [54,57].

The active thermal range of syrphids, while species-specific, is impacted by thermal reflective properties of the insect cuticle, along with body size and behaviour, with a negative relationship existing between flight activity and thoracic width [22,175]. Larger taxa typically achieve and sustain body temperatures required for flight more readily than smaller taxa, and are thus more active in the early morning and evening [128]. Hairier, smaller and more reflective taxa are better suited to maintaining thermal balance at higher temperatures during flight, and are thus active in the middle of the day when temperatures peak. In comparison, the flight activity of larger, less reflective (darker) taxa declines with increasing temperature, and foraging during the peak of the day can be restricted to the shade [22,128]. 

Syrphid activity is impacted by cloud cover, wind and rain, seemingly to a greater extent than for most blow flies (which are generally larger in body size). Regardless, *E. tenax* actively forages at temperatures as low as 5 °C [58]. Temporal variation in foraging activity will occur across species and families and, for potential pollinating agents, must be assessed alongside the timing of target crop flowering. Access to shade during the hotter periods of the day was noted as an important requirement for retention in the release area for *S. grandicornis,* suggesting that manipulation of site factors could improve pollination abilities of target taxa [175]. Additionally, resource requirements may influence an individual’s response to floral cues, such as odour, with sexual dimorphism in resource needs potentially accounting for a proportion of the temporal variation noted in foraging activity within syrphid species [22,128]. 

Nectar foragers will obtain resources to sustain flight and extend lifespan, while pollen feeding supports the reproduction requirements of both sexes [196,201]. Sexual dimorphism was not evident in flower visitation at the family level of syrphid taxa, although differences in foraging requirements (pollen vs. nectar feeding) between sexes have not been studied and likely would be evident, depending on the physiological status of the individual [173]. Females, likely requiring protein for egg development, have been reported to feed on pollen more frequently than males [196]. Males are usually more active than females, potentially resulting in higher sugar requirements and more frequent nectar feeding [173]. 

In general, hover flies are preferentially attracted to white or yellow flowers, representing a long range visual cue during foraging [202,203]. Klecka et al. (2018) reported that *Ep. balteatus* was less selective towards flower colour than species from other subfamilies. No differences in specialisation and frequency of flower visitation were noted between males and females. Differences in colour preferences likely reflect foraging preferences rather than differences in the visual system between species [173]. Further research is needed to explore other cues used in foraging, including the role of semiochemcials and UV reflectance. Hover flies use olfactory cues to locate suitable oviposition sites, but little is known regarding the importance of olfactory cues in nectar and pollen foraging [204].

### 8.3. Rearing and Release

Life history data specific to Australian syrphid taxa are largely lacking, although biological requirements are diverse across genera, with both predacious larvae and semi-aquatic breeding substrates reported [175]. For many syrphid species, successful rearing will require the additional rearing of a second insect taxon (e.g., aphids) as a food source [175], which will increase the cost and complexity of rearing protocols [172]. Selection of taxa that are not reliant on a secondary insect food source for development, such as *E. tenax*, is therefore preferable when determining suitable candidates for pollination services. 

Rearing protocols have been established for *E. tenax* with adults maintained on nectar and pollen, while the larvae can be successfully reared on a diet of decomposing organic matter such as cow manure [58,172]. However, the larvae are aquatic, breeding in liquid substrates of decaying organic matter, including wet decaying seaweed, stagnant water bodies polluted with manures, human sewage and decomposing animal carcasses [58]. The posterior spiracles of the aquatic larvae of *E. tenax* are located on a retractable caudal segment that, extended (up to three times the length of the larvae), provides a breathing tube allowing larvae to obtain air from the surface of a water body while submerged. Due to their appearance, the larvae are commonly called ‘rat-tailed’ maggots [205]. As larvae require semi-aquatic substrates for development, mass rearing efforts can be more complex than those required for calliphorid taxa. Furthermore, in laboratory rearing, first instar larval mortality is significant and may limit effective mass production for commercial release, unless further development of *E. tenax* rearing protocols reduces developmental mortality [171]. The development of optimal rearing protocols is thus necessary to maintain syrphids in colony and minimise the costs associated with mass rearing species for release. 

As with calliphorids, the development of syrphids can be manipulated by temperature control to facilitate the synchronisation of pupation and emergence with transport and application timing [206]. The development of *E. tenax* can be optimised under temperature control, with egg to adult development completed within 24–36 days at 21.5 °C or 22.65 ± 0.189 at 25 °C [58,171,172]. Reproductive output is high, with lifetime fecundity reported at 3,000 eggs/female [172]. Inducible hibernation of adult *E. tenax* can be achieved for prolonged periods under conditions of complete darkness and low temperatures of 8–10 °C, providing hibernation is disrupted every 3–4 days to allow grooming and feeding to occur before returning the adults to a hibernation state [172]. Inducible adult diapause options are likely to minimise colony maintenance inputs during periods when demand for pollination services is low. Inducible pupal diapause is also reported in syrphids in response to environmental conditions (photoperiod and temperature), facilitating the potential to optimise ‘pupal shelf life’ for commercial distribution in line with calliphorid taxa [207,208].

Upon release, syrphids such as *E. tenax* are capable of travelling up to 75 km a day, but foraging opportunities and physiological status direct flight activity and dispersal [58]. The adult life span upon release is dependent on species, temperature and nutritional intake (Table 2). The adult lifespan of *Ep. baleatus* ranges from 7–13 days depending on diet and levels of sucrose, glucose and fructose in nectar resources [201,209]. In contrast, the adult lifespan of *E. tenax* in greenhouse releases (temperature uncontrolled) is between 2–4 weeks, while adults under controlled temperatures of 21.5 °C can survive for up to 3 months [57,172]. Similarly, adult longevity of *S. grandicornis* ranges between 13–30 days [175].

Unlike the majority of syrphids, *E. tenax* is considered a minor nuisance fly with larvae reported to infest livestock food sources, effluent and other partial liquid sources, including association with, and disruption to, dairy farming equipment. The species has been reported to cause human myiasis in extremely rare cases [58]. Despite its minor pest status and rearing complexity in comparison with calliphorids, *E. tenax,* along with other syrphids, has considerable potential as a managed pollinator providing oviposition issues are mitigated by the release of sterile adults (in field situations), and rearing protocols are optimised for speed and cost efficiency. Further research regarding the life cycle and life history traits of identified syrphid species is necessary to develop successful mass rearing protocols, optimise stocking rates, explore sterile insect technique approaches and to determine the timing and required number of releases throughout the target crop pollination period.

## 9. The Future of Managed Flies for Crop Pollination

While this review has identified species demonstrating potential for development for commercial pollination services (Table 2), data pertaining to their pollination efficiency, effectiveness and practical implementation as crop-specific managed pollinators are largely lacking. A subsequent research objective building on the identification of potential fly pollinators is to assess the pollination abilities of individual species, particularly in terms of impact on yield and quality of fruit or seed in target crops. Where viable species are determined, further research can then be directed towards knowledge gaps regarding life history parameters in these taxa, in order to develop and optimise procedures for mass rearing, management and the implementation of pollination services in the horticultural industry. Specific areas requiring further investigation include fly foraging behaviour in relation to temperature, photoperiod and solar radiation, visual and odour cues eliciting an attraction response and nutritional requirements driving flower visitation and dispersal. 

The development of fly pollinators for horticulture will also require assessments of relevant life history traits in candidate species to determine their suitability for rearing in large numbers. Such assessments would need to consider reproductive mode (i.e., oviparous or ovoviviparous); ability to be mass-reared, including the number of days between emergence, mating and oviposition and larval and pupal development rates; female lifetime fecundity; dispersal rates in the field; lifespan in the field or protected cropping conditions; species seasonality and/or temperature ranges at which they are actively flying and feeding on flowers; and potential to negatively impact upon humans, livestock or wild pollinator biodiversity.

Projects employing several different fly species to convert organic wastes into larval and pupal byproducts provide some insights into the challenges involved with rearing flies on a commercial scale [210]. Dipterans such as the black soldier fly, *Hermetia illuscens* Linnaeus 1758 (Stratiomyidae), the house fly (*M. domestica*) and several blowflies, including *Ch. megacephala,* are already being used on a large scale to reduce bio-waste streams [211,212].

## 10. Commercial Pollination Services

Outside of Australia, we know of only three companies that produce flies for pollination services when all accessed on 28th May, 2020: (1) Poly Fly (Almeria, Spain) provide the hover fly *Eristalinus aeneus* Scopoli 1763 (Syrphidae) in Europe to pollinate seed crops (brassicas), vegetables (cucurbits and solanaceous) and fruits (e.g., avocado) (www.polyfly.es); (2) Koppert Biological Systems (The Netherlands) supply the blow fly, *L. sericata* (Calliphoridae), for pollination of isolated crops on a small scale such as seed crops (www.koppert.com/natupol-fly/); and (3) Forked Tree Ranch (Idaho, US) supply the bluebottle fly, *Calliphora vomitoria* Linnaeus 1758 (Calliphoridae), for pollination of vegetable crops (e.g., onions, peppers, carrots, brassicas, radish, garlic, lettuce) as well as buckwheat, canola and sunflowers. In these applications, the flies are primarily released outdoors in tents ranging from selfing cages to 30 m tents/hoops and tunnels (www.forkedtreeranch.com). A fourth company Muxidotecnia based in Chile provided flies for seed crop pollination, with website graphics depicting both *L. sericata* and *Ch. megacephala*, (Calliphoridae) but providing very little other information (www.muxidotecnia.cl) (last accessed on 12th July, 2018 but no longer accessible). Within Australia, the only known provider of flies as pollinators is Sheldon’s Bait (www.sheldonsbait.com.au), who advertise that they “can provide ready to hatch blow fly pupae for your pollination needs”(accessed on 28th may, 2020).

## 11. Conclusions

Considering current service providers and the existing pollination evidence around flies, we have identified 20 fly taxa from three key families (Calliphoridae, Rhiniidae and Syrphidae). The identified species demonstrate biological characteristics aligned to traits associated with pollination success and thus suitability for further development as managed pollinators within an Australian context (Table 2). Flies are important crop pollinators with the potential to compliment bee pollination, both as a managed pollination service or in the management of existing wild pollinators. The collative approach outlined to identify suitable fly taxa is applicable to the identification of managed pollinators in other cropping systems and regions of the world. Many of the species identified in an Australian content have a global distribution and will likely prove suitable candidates in other localities and geographic regions. Selection of taxa at a local level is mandated to ensure minimal impact to existing pollinators and local biodiversity. The outlined guide to candidate identification, and the species identified in this Australian case review, should drive research efforts to further advance our understating of fly pollination abilities and the development of optimal application processes for the implementation of such taxa as pollination agents globally.

## Figures and Tables

**Figure 1 insects-11-00341-f001:**
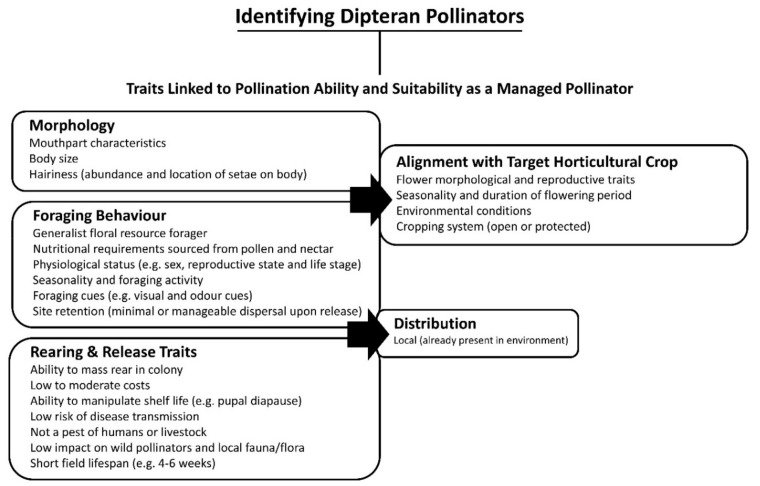
A proposed framework outlining relevant species-specific trait assessment criteria necessary for the identification of potential dipteran pollinators in a horticultural crop setting.

**Figure 2 insects-11-00341-f002:**
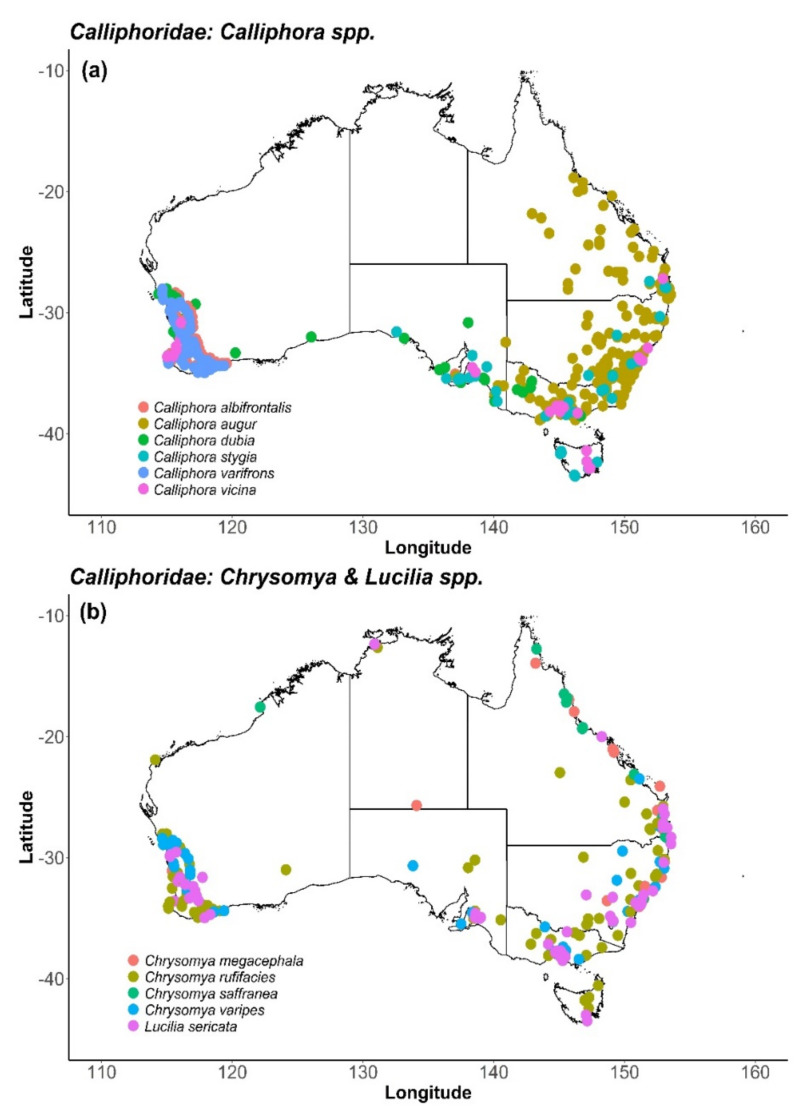
Known distribution of Calliphoridae taxa, (**a**) genus *Calliphora*; (**b**) genus *Chrysomya* and *Lucilia*, identified as potential pollinators of horticultural crops in Australia. Records sourced from Atlas of Living Australia [131].

**Figure 3 insects-11-00341-f003:**
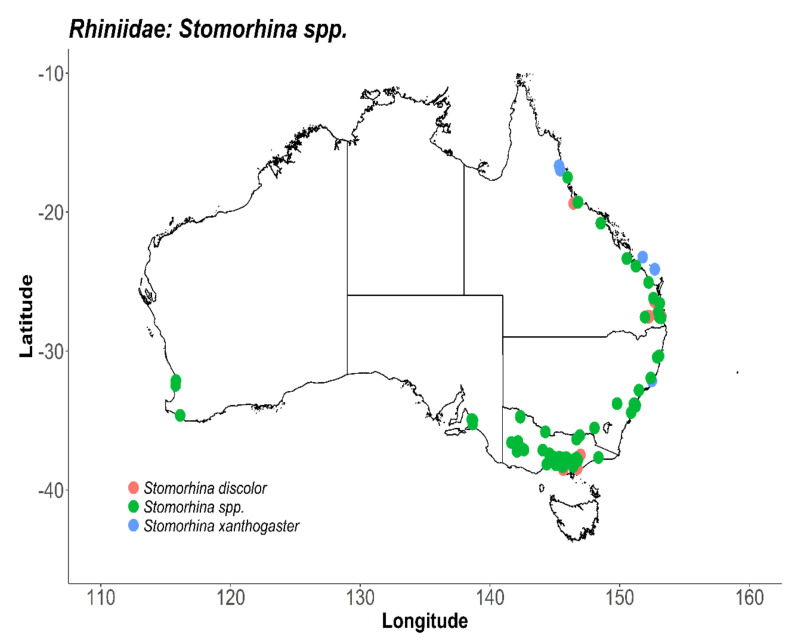
Known distribution of Rhiniidae taxa within the genus *Stomorhina*, identified as potential pollinators of horticultural crops in Australia. Records sourced from Atlas of Living Australia [131].

**Figure 4 insects-11-00341-f004:**
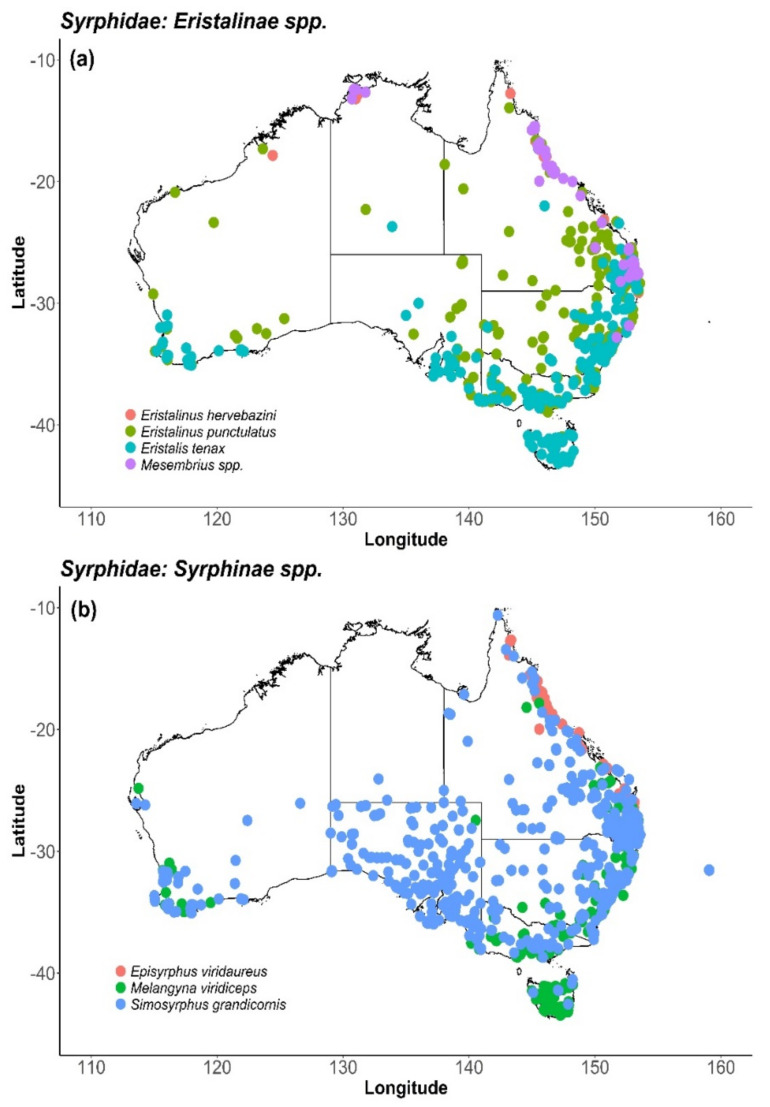
Known distribution of Syrphidae taxa in the subfamilies (**a**) Eristalinae; and (**b**) Syrphinae, identified as potential pollinators of horticultural crops in Australia. Records sourced from Atlas of Living Australia [131].

**Table 1 insects-11-00341-t001:** Fly species visiting flowers and/or pollinating crops relevant to Australian horticulture. Literature detailing pollination metrics of efficiency and/or effectiveness are in **bold** text.

Fly Pollinator	Crops ^A^	Literature
**Anthomyiidae (Flower Flies)**		
*Delia* spp.	PK	**[16]**
*Delia platura*	PK	[14,27,28]
*Anthomyia punctipennis*	CT, ON, PK	[14], **[16]**, [27]
**Bombyliidae (Bee Flies)**		
*Comptosia ocellata*	CT	[29]
**Calliphoridae (Blow Flies)**	AV, BL, BS, CT, MC, MG, ON	[29,30,31,32,33,34,35,36,37,38]
*Calliphora* spp.	AV, MC, MG, ON	[35,36,37]
*Calliphora vicina*	AV, CT, LC, LK, ON, PK	[14,27,39,40]
*Calliphora stygia*	AV, ON, PK	[14], **[16]**, [27], **[32]**, [41]
*Calliphora augur*	AV	**[32]**, **[37]**
*Calliphora albifrontalis*	AV, BL	[42]
*Chrysomya* spp.	MG	[37,43,44]
*Chrysomya megacephala*	MG	[43,45,46,47]
*Chrysomya saffranea*	MG	[43]
*Chrysomya rufifacies*	AV, MG	[32,43]
*Chrysomya varipes*	AV	**[36]**
*Lucilia sericata*	AV, CE, LY, MG, ON, PK, ST	[14,27,32,48,49,50]
*Lucilia cuprina*	AV	[32]
**Muscidae (House Flies)**		[29,37,51]
*Musca domestica*	LK, MG, ON	[27], **[35]**, [39], **[43]**, **[50]**
*Hydrotaea rostrata*	ON, PK	[14,27]
*Spilogona* spp.	PK	[28]
**Rhiniidae (Nose Flies)**	AV	[32,37]
*Stomorhina discolour*	AV, MC, MG	[32], **[37]**
*Stomorhina xanthogaster*	MG	[43]
**Sarcophagidae (Flesh Flies)**	MG, ON	[35,37,43]
*Oxysarcodexia varia*	ON, PK	[14], **[16]**, [27]
**Stratiomyidae (Soldier Flies)**		[37,52]
*Odontomyia* spp.	PK	[14,27,28]
*Odontomyia atrovirens*	PK	[52]
**Syrphidae (Hover Flies)**	AV, LY, MC, MG, ON, ST	[10,13,14,28,37,47,52,53,54,55]
*Eristalis* spp (Drone Flies)	MG	[43,44]
*Eristalinus hervebazini* ^B^	MG	**[43]**
*Eristalis tenax*	CB, CT, ON, PE, PK	[14,15,16,27,28,29,38,41,51,55,56,57,58]
*Melangyna* spp.	CT, ON, PK	[14,27,29,51]
*Mesembrius bengalensis*	MG	**[43]**
*Simosyrphus grandicornis*	AV	**[32]**
**Tachinidae (Bristle Flies)**	AV, ON, PK	[14,27,32,37,43,52]
**Tabanidae (Horse Flies)**		
*Scaptia* spp.	PK ON	[14,27]

^A^ Avocado (AV); Blueberry (BL); Brussels Sprout (BS); Carrot (CT); Celery (CE); Cranberry (CB); Leek (LK); Lucerne (LC); Lychee (LY); Macadamia (MC); Mango (MG); Onion (ON); Pak Choi (PK); Peppers (PE) and Strawberry (ST). ^B^ Synonym *Eristalis maculatus.*

**Table 2 insects-11-00341-t002:** Biology and life history traits of fly species occurring in Australia identified as potential commercial pollinators of horticultural crops in Australia.

Fly Pollinator	Size (mm)	Eggs/♀	Lifespan	DT	DP	ATR	Literature
**Calliphoridae (Blow flies)**							
*Calliphora vicina*	10–14	111–250	40 (F)	15		4–25	[132,133,134,135,136]
*Calliphora stygia* **(N)**	8–13	85	35 (L)	18	10	7–24	[136,137,138,139,140,141,142]
*Calliphora augur^VIV^* **(N)**	11		67 (F)	13	8	11–26	[139,140,143,144,145]
*Calliphora dubia^VIV^* **(N)**	11	50–80		19			[145,146,147,148]
*Calliphora albifrontalis* **(N)**				20			[146]
*Calliphora varifrons^VIV^* **(N)**		33.4 ± 1.0		19	12	6–32	[146,149,150]
*Chrysomya megacephala*	8–9	223.7 ± 2.4	54–105 (L)	19	6		[124,140,151,152,153,154,155,156,157]
*Chrysomya rufifacies* **(N)**	9	210	24 (L)	12	5	13–28	[139,140,145,146,158,159,160]
*Chrysomya saffranea* **(N)**				12		18–33	[140,157,161]
*Chrysomya varipes* **(N)**	6			11	5	20–28	[140,162]
*Lucilia sericata*	10–14	225 ± 7	30 (F) 40–59 (L)	13	6	10–30	[139,163,164,165,166,167,168,169]
**Rhiniidae (Nose Flies)**							
*Stomorhina discolour*	4.5–6.5						[170]
*Stomorhina xanthogaster*	9.5–11						[170]
**Syrphidae (Hover Flies)**							
*Eristalinus hervebazini* **(N)**							
*Eristalinus punctulatus* **(N)**							
*Eristalis tenax*	14.52	191.4 ± 75.7	23.4 ± 2.4 (L)	22.6 ± 0.2		5–30	[171,172,173]
*Episyrphus viridaureus*		45.0 ± 16.8	14.0 ± 1.5 (L)	29–30	7		[174]
*Melangyna viridiceps* **(N)**		288.0 ± 34.0	32.9 ± 1.6 (L)	17.8	7.1		[175,176]
*Mesembrius bengalensis*							
*Simosyrphus grandicornis*		307.9 ± 23.2	19.9 ± 1.4 (L)	17.4	8.2		[175,176]

***^VIV^*** = lay live larvae; **(N)** = native species; Size = body length (mm) as a measure of body size; Eggs/♀ = eggs or larvae laid per female; **Lifespan** = adult longevity (days) [(F) Field,(L) Laboratory]; **DT** = development time at 25 °C from egg to adult emergence (days); **DP** = duration of pupal stage at 25 °C; and **ATR** = active thermal range (°C), e.g., foraging/flying adults.

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
