# Peer review of "The Role of Flies as Pollinators of Horticultural Crops: An Australian Case Study with Worldwide Relevance"

_insects, 2020, doi:10.3390/insects11060341_

Round 1

Reviewer 1 Report

This is a comprehensive review of the potential Dipteran pollinators in Australia. I think it will be useful as a compendium of relevant literature and what is currently known. The illustrations are clear, and the table is useful. 

My major complaint is that the authors should have read it more carefully for editorial issues before submitting it, as it makes it more tedious to review if there are many minor grammatical issues or misspellings to correct.  I've marked a lot of corrections on the PDF.  

Author Response

Reviewer #1: This is a comprehensive review of the potential Dipteran pollinators in Australia. I think it will be useful as a compendium of relevant literature and what is currently known. The illustrations are clear, and the table is useful. My major complaint is that the authors should have read it more carefully for editorial issues before submitting it, as it makes it more tedious to review if there are many minor grammatical issues or misspellings to correct.  I've marked a lot of corrections on the PDF.  

Response: We have corrected and addressed each of the minor grammatical issues and/or misspellings throughout the manuscript as directed (indicated in track changes). Specific edits in response to reviewer 1’s comments are detailed below.

Line 128 - All native?

Changed ‘individual species’ to ‘introduced and endemic species’ to indicate the review scope was not limited to native fly species occurring in Australia.

Line 110 (Table 1 caption) - Need to underline?

Underlined caption legend term ‘bold’ to indicate literature detailing pollination metrics of efficiency and/or effectiveness are in bold and underlined text.

Line 146 - Is it native there, or introduced?

Added ‘cosmopolitan’ to indicate the species has a wide geographic distribution and is not endemic to Australia. Native status now also indicated in Table 2.

Line 153 - Also shown in Kearns and Inouye (1994); compared to what?

Revised sentence by moving citations to the end of the sentence to added suggested reference ‘Kearns and Inouye (1994)’ and added ‘than bees’ to indicate comparison species as requested.

Line 222 - What?

Removed contradictory sentence to clarify earlier statement ‘Despite this, attempts to enhance blueberry pollination with supplemental bees have proven unsuccessful to date’.

Line 368 - How?

Revised sentence to indicate how bees and flies differ in foraging behavior adding ‘…may be comparatively less likely to cause pollination compared with honeybees. However, both taxa actively collect or eat pollen/nectar and also carry loose pollen on their body. Some solitary bees may have a lot of loose body pollen, but honeybees and stingless bees carry most food pollen in sticky pollen balls that do not contribute to pollination. …’.

Line 492 (Table 2) - All native?

We now indicate native species where known in Table 2 using the code (N) with accompanying footnote. All Table 2 species occur in Australia but records are limited and we cannot accurately identify original origins of all species where there are no records on introduction and they occur widely or throughout the Australasian region.

Spell this out. I don't know what SIT is.

Line 690, the acronym ‘SIT’ is defined earlier but we have now removed the use of the acronym throughout the manuscript (line 690, line 733 and line 905).

Might be worth citing: Rotheray, G. E. and F. Gilbert (2011).

Page 18, as we have not specifically referred to the suggested reference we won’t cite it in this instance but appreciate the suggestion.

Reviewer 2 Report

This is a very interesting review, and the authors have compiled a large amount of biological and ecological information on a topic which deserves more attention. I enjoyed reading it.

Good general introduction, however I thought several topics should have been highlighted further

  • Invasion risk and considering native pollinators first
  • Disease risk from using commercially reared pollinators
  • Management techniques to support wild pollinator populations in situ rather than commercial rearing and release. While I appreciate this is a review of commercially managed pollinators, a brief paragraph highlighting the possibility of in situ management and why in some cases commercial management of pollinators is necessary would be informative.

To meet the aim of this becoming a road map, I would suggest a summary table of the kind of questions that need to be answered, or a flow diagram – for example what do you need to know about your target crop? What would you need to know about the potential pollinators? And what do you need to cross reference between the two? I would also like to see the issues around risk of using around using commercially bred pollinators highlighted much more.

Related to the comment above, some structural re-working would be helpful – I sometimes found the text to be jumping between topics, in particular general information was sometimes in the specific information section and vice versa for example –

Line 254 – while talking about raspberries and other fruit, there is a paragraph about foraging and thermal tolerance in different insects. While it is an important point that the crops grown may be covered and therefore in different conditions to outdoor crops and the information on conditions should be kept, the information on different groups, then belongs under specifics for those taxon.

Line 353/Line 426 – both sections talk about pollinator effectiveness. The paragraph starting at Line 353 I think would be better expanded, including definitions of SVD and more information on general pollinator characteristics – in fact most of the information in the paragraph starting at line 426. Then in line 426 you could focus on how blowflies fit/don’t fit the framework you have already outlined. At present there is some general information combined with the taxon specific information.

Line 558 and the two paragraphs following – this is very interesting information about how reproductive strategies and life stage might affect pollination efficiency, which I think it would be important to consider across all groups of flies, so should be highlighted in the paragraph at line 370.

Minor comments

Line 24 ‘data available are limited’ not ‘data available is limited’

Line 38-40 First two sentences are repeated

Line 70 – I think it worth highlighting the particular issues for australian agriculture, including a brief discussion of are many of these crops exotic and not close to native Australian species, and how might this impact their pollination?

Line 82 – what was the rationale for these particular crops being selected? More information appears at line 114, but a brief sentence about the importance of these crops to Australian horticulture would be helpful for a global audience.

Line 320– You follow on straight from talking about carrion to encourage flies in fields to a hoverfly – potentially misleading in that they have quite different life histories so it would be worth giving brief detail about these –

Line 330  for example, what are the life history traits associated with E. tenax that are positive

Line 406 – where honey bees are yet to be introduced – presumably this is a very small area

Line 426 – single visit deposition – this is actually not the correct paper for the definition Winfree et al 2007 https://doi.org/10.1111/j.1461-0248.2007.01110.x

Table 2 –include a column for native range

Ref 38 – I assume this is a thesis? The full citation should follow the Insects guidelines

Thesis:

8. Author 1, A.B. Title of Thesis. Level of Thesis, Degree-Granting University, Location of University, Date of Completion

Author Response

Reviewer #2: This is a very interesting review, and the authors have compiled a large amount of biological and ecological information on a topic which deserves more attention. I enjoyed reading it.

Good general introduction, however I thought several topics should have been highlighted further

  • Invasion risk and considering native pollinators first
  • Disease risk from using commercially reared pollinators
  • Management techniques to support wild pollinator populations in situ rather than commercial rearing and release. While I appreciate this is a review of commercially managed pollinators, a brief paragraph highlighting the possibility of in situ management and why in some cases commercial management of pollinators is necessary would be informative.

Response: The issue of invasion risk and impact on native pollinators is currently addressed in the introduction where we state, ‘Regionally specific assessment of suitable taxa is a necessary process to avoid unwanted / pest interactions with other pollinators and locally occurring biodiversity.’ and feel that further detail on this aspect would unnecessarily extend an already long introduction.

We have inserted the line to ‘and present negligible risk of disease transmission to existing wild pollinators when reared under screened colony conditions’ to the introduction to highlight the issue of potential disease transmission.

Rather than add to the introduction length and detract from the focus on commercially managed pollinators with an entire paragraph detailing the potential to facilitate wild pollinators, we have included a succinct statement ‘The identification of potential fly pollinators offers further value in directing research efforts around management techniques that facilitate wild pollinator populations in addition to the development of managed pollination services.’ to address this aspect.

To meet the aim of this becoming a road map, I would suggest a summary table of the kind of questions that need to be answered, or a flow diagram – for example what do you need to know about your target crop? What would you need to know about the potential pollinators? And what do you need to cross reference between the two? I would also like to see the issues around risk of using around using commercially bred pollinators highlighted much more. 

Response: We agreed this would be a valuable addition and have now included a flowchart detailing the framework for identification of potential pollinator taxa (now Figure 1). We again mention risks associated with use of commercial pollinators in this figure to highlight this component of assessment.

Some structural re-working would be helpful – I sometimes found the text to be jumping between topics, in particular general information was sometimes in the specific information section and vice versa for example –

Line 254 – while talking about raspberries and other fruit, there is a paragraph about foraging and thermal tolerance in different insects. While it is an important point that the crops grown may be covered and therefore in different conditions to outdoor crops and the information on conditions should be kept, the information on different groups, then belongs under specifics for those taxon.

Response: Agree and the paragraph relating to foraging and thermal tolerance was moved out of the inappropriate ‘raspberry section’ and reworked into the ‘Calliphoridae distribution and foraging section’. The paragraph was also shortened by removing the repeated detail on the active thermal range of bees and C. vicina.

Line 353/Line 426 – both sections talk about pollinator effectiveness. The paragraph starting at Line 353 I think would be better expanded, including definitions of SVD and more information on general pollinator characteristics – in fact most of the information in the paragraph starting at line 426. Then in line 426 you could focus on how blowflies fit/don’t fit the framework you have already outlined. At present there is some general information combined with the taxon specific information.

Response: Agree and we have moved the discussion of single visit deposition and pollinator characteristics from the specific ‘Calliphoridae morphology’ section to the general introduction of Pollinators and removed ‘Pollen load has been used as a predictor of SVD in conjunction with behavioural and morphological traits that determine the insect-flower interaction.” to shorten length.

Line 558 and the two paragraphs following – this is very interesting information about how reproductive strategies and life stage might affect pollination efficiency, which I think it would be important to consider across all groups of flies, so should be highlighted in the paragraph at line 370.

Response: We have highlighted this aspect where indicated by inserting ‘Sex, reproductive status and/or life stage can influence foraging ecology in response to fluctuating nutritional requirements and should be considered when assessing the synchrony of crop and potential pollinating taxa’.

Line 24 ‘data available are limited’ not ‘data available is limited’

Response: Amended.

Line 38-40 First two sentences are repeated

Response: The copy error has been amended by deleting the repeating lines. 

Line 70 – I think it worth highlighting the particular issues for australian agriculture, including a brief discussion of are many of these crops exotic and not close to native Australian species, and how might this impact their pollination? 

Response: All horticultural crops discussed are exotic to Australia and there are no data relating to native fly foraging on endemic crops or plants to allow lengthy discussion on this issue. However, we acknowledge the issue by stating ‘While almost all horticultural crops are exotic to Australia (macadamia is one such exception), such crops likely represent an additional source of nutrition to both native and exotic fly species that are known generalist foragers.’ As detailed in the manuscript, a requirement of identification of potential fly species is that there is evidence of flower visitation and/or pollination of these exotic crops which ensures the selection of generalist foragers attending the exotic crops.

Line 82 – what was the rationale for these particular crops being selected? More information appears at line 114, but a brief sentence about the importance of these crops to Australian horticulture would e helpful for a global audience.

Response: While we appreciate the section detailing ‘search terms’ for the review does not include detail on crop selection we feel this is clearly stated in the first two paragraphs under ‘Target Horticultural Crops’. However, we have now added the sentence ‘ Crops of interest were selected based on common Australian horticultural produce, known requirement for insect pollination to improve yield and to represent a range of cropping systems and environments.’ under the ‘Search Strategy and Selection Criteria’ section as well.

Line 320– You follow on straight from talking about carrion to encourage flies in fields to a hoverfly – potentially misleading in that they have quite different life histories so it would be worth giving brief detail about these –

Line 330 for example, what are the life history traits associated with E. tenax that are positive

Response: We have revised the sentence to clarify that E. tenax does not breed in carrion by stating ‘Some evidence exists relating to the non-carrion breeding drone fly, Eristalis tenax…’ and added the following to positive detail on life history traits of the drone fly, ‘including rearing in cheap and easily obtained substrates…’ with further life history detail already provided later in the review.

Line 406 – where honey bees are yet to be introduced – presumably this is a very small area

Response: We have revised the sentence to clarify the importance of flies in relation to bees to read, ‘Alongside both managed and wild bees, blow flies are likely the main crop pollinating insect.’

Line 426 – single visit deposition – this is actually not the correct paper for the definition Winfree et al 2007 https://doi.org/10.1111/j.1461-0248.2007.01110.x

Response: We have included the indicated reference to the statement in addition to the existing citation to support the concepts reported.

Table 2 –include a column for native range

Response: We have now addressed this (see response under Reviewer 1).

Ref 38 – I assume this is a thesis? The full citation should follow the Insects guidelines

Thesis:

  1. Author 1, A.B. Title of Thesis. Level of Thesis, Degree-Granting University, Location of University, Date of Completion

Response: The reference has been reformatted to match guidelines ‘Spurr, C. Identification and management of factors limiting hybrid carrot seed production in Australia. PhD Thesis, University of Tasmania, Tasmania, 2003.’

Reviewer 3 Report

The authors have produced a thorough literature review on the role of flies as pollinators of Australian horticultural crops. I think the study is interesting, of global relevance and is worthy of publication. However, I do have a few (minor) comments.

Firstly, this review is pretty long and at some points overly wordy (Insects guide to authors for reviews: “concise and precise updates”). I recommend the authors take a serious look at their subheadings and check that the content therein actually applies. There’s a few places where you introduce very general concepts e.g. single visit deposition, breeding on carrion, and temperature ranges, under subheadings which I think would be better suited to the introduction. I would then only include the species or crop specific information under the subheadings. Regarding the wordiness there’s certainly a few places which could be cut-down, e.g. 514-516 and 545-547 are essentially repetition, and I recommend you thoroughly check this manuscript for similar occurrences.

Secondly, since your horticultural crops occur in open and closed cropping systems, I think you need to be careful with your use of language and depth of discussion. For example, in figures 1, 2 and 3 you refer to flies as ‘alternative’ pollinators, but this is troublesome in open-field systems because flies will likely compliment bees (you mention this briefly around 310-315), and managed pollinators will likely compliment wild pollinators. I would therefore be tempted to remove the word ‘alternative’ in this context and would broaden your conclusion slightly to say that overall, flies are great pollinators and that they can be complimentary to bees, regardless of whether or not they are specifically introduced.

197 – This seems too speculative - did these studies not look at Rhinidae species at all?

215 – or that honeybees are inefficient in this context.

219 – reference needed for blueberry pollination requirements.

250 – reference needed

278 - reference needed

288 – replace ‘male sterile’ to ‘female seed-bearing’ or similar

293 – you mention here and a few other places that bees can be distracted by alternative forage to the target crop. Do we know any similar information about flies which could balance this statement?

336 – this sentence is too long and needs to be reworded

355- morphological (and physiological?) traits such as mouthparts ….

369 – what conditions?

371 – what about warmer temperatures? Surely this is likely in Australia and/or in closes systems and/or under climate change etc.

476 – reference needed

582 – (last clause) why? I don’t think this is well explained in the reference either

Figures 1,2,3 – is there a reason they are only recorded in these regions? Is it sampling effort, climatic suitability? Also, where are these crops produced in Australia? Is there potential for a spatial mismatch of production and native species? You briefly mention this 518-520 but its lost under the distribution and foraging behaviour of Calliphoridae.

Author Response

Insects-795107: Response to Reviewer Comments

We thank the reviewers for their suggested comments and revisions which we have now implemented to further refine and improve the manuscript.

Reviewer #3: The authors have produced a thorough literature review on the role of flies as pollinators of Australian horticultural crops. I think the study is interesting, of global relevance and is worthy of publication. However, I do have a few (minor) comments.

Firstly, this review is pretty long and at some points overly wordy (Insects guide to authors for reviews: “concise and precise updates”). I recommend the authors take a serious look at their subheadings and check that the content therein actually applies. There’s a few places where you introduce very general concepts e.g. single visit deposition, breeding on carrion, and temperature ranges, under subheadings which I think would be better suited to the introduction. I would then only include the species or crop specific information under the subheadings. Regarding the wordiness there’s certainly a few places which could be cut-down, e.g. 514-516 and 545-547 are essentially repetition, and I recommend you thoroughly check this manuscript for similar occurrences.

Response: We agree there are a few areas that can be revised to reduce the length of the document and focus subheading content as directed by both Reviewer 2 and 3. We have completed a final check throughout the manuscript for revision of sections to provide a ‘more focused and concise’ review. Shifted content is detailed under Review 2 comments and additional revisions are detailed below.

Original Line 547, deleted repetitive sentence ‘Temperature and radiation intensity were likely the most important environmental factors influencing flight activity in this study.’

Original Line 560, shortened the sentence indicated to avoid repetition by removing ‘differ considerably to other taxa, including bees, and likely result in different spatial and temporal activity patterns that...’

Original Line 564, revised the sentence to be more concise.

Secondly, since your horticultural crops occur in open and closed cropping systems, I think you need to be careful with your use of language and depth of discussion. For example, in figures 1, 2 and 3 you refer to flies as ‘alternative’ pollinators, but this is troublesome in open-field systems because flies will likely compliment bees (you mention this briefly around 310-315), and managed pollinators will likely compliment wild pollinators. I would therefore be tempted to remove the word ‘alternative’ in this context and would broaden your conclusion slightly to say that overall, flies are great pollinators and that they can be complimentary to bees, regardless of whether or not they are specifically introduced.

Response: We agree that fly pollination, while an alternative to bees is in some cases more likely to augment bee pollination. We have rephrased the wording throughout the manuscript to ensure this is clarified by removing the term ‘alternative’ or where retained, adding ‘supplement’, as detailed in the track changes. Additionally we have expanded the conclusion as directed by inserting ‘Flies are important crop pollinators with the potential to compliment bee pollination, both as a managed pollination service or in the management of existing wild pollinators.’

197 – This seems too speculative - did these studies not look at Rhinidae species at all?

Response: Agree and we have removed the speculative sentence.

215 – or that honeybees are inefficient in this context.

Response: We have amended the sentence to now indicate this aspect by inserting ‘honey bees may be inefficient in this context and/or’.

219 – reference needed for blueberry pollination requirements.

Response: Added reference Danka et al (2019) to support statement.

250 – reference needed

Response: Added reference Madrid & Beaudry (2020) to support statement.

278 - reference needed

Response: Added reference Spurr (2003) to support statement (Line 281) and amended sentence to read ‘flies can be used’ instead of ‘are often used’.

288 – replace ‘male sterile’ to ‘female seed-bearing’ or similar

Response: This suggested change is incorrect terminology and when revised the sentence does not make sense. To ensure hybridisation, the female (or hybrid seed bearing line) used in hybrid seed crops is bred to be male sterile. This is conferred by a group of cytoplasmic genes that cause anther abortion or substitution of the anthers with petal like structures. The accepted term for a hybrid seed parent line displaying these traits both in scientific and commercial terminology is male sterile or cytoplasmically male sterile – these are used interchangeably. The seed that is borne by the female line is not female seed as the revision implies.

293 – you mention here and a few other places that bees can be distracted by alternative forage to the target crop. Do we know any similar information about flies which could balance this statement?

Response: Unfortunately no but we do indicate in our review that this is an area requiring further research. No changes made.

336 – this sentence is too long and needs to be reworded

Response: The sentence has now been split into two sentences to read ‘Strategies to retain fly pollinators in open field crops are needed to avoid unintended negative consequences of mass fly releases on surrounding land users and the environment. Additionally, the development of flies as alternative pollinators of vegetable seed crops will also require innovation of pest control practices and/or pollinator species selection and release strategies to mitigate potential losses incurred through pesticide application.’.

355- morphological (and physiological?) traits such as mouthparts ….

Response: Agreed and ‘physiology’ has been added to acknowledge this.

369 – what conditions?

Response: We have removed the words ‘for a given set of conditions’ as superfluous following revision of the sentence under Review 1 responses.

371 – what about warmer temperatures? Surely this is likely in Australia and/or in closes systems and/or under climate change etc.

Response: Yes true, as dipteran thermal tolerance is generally broader than bees in both directions. We have amended the statement to read ‘e.g. remain active over a wider range of temperatures’.

476 – reference needed

Response: Added reference Smith et al (2015) to support statement.

582 – (last clause) why? I don’t think this is well explained in the reference either

Response: We have revised the sentence to clarify this aspect and added ‘due in part to the comparatively longer period of gestation’.

Figures 1,2,3 – is there a reason they are only recorded in these regions? Is it sampling effort, climatic suitability? Also, where are these crops produced in Australia? Is there potential for a spatial mismatch of production and native species? You briefly mention this 518-520 but its lost under the distribution and foraging behaviour of Calliphoridae.

Response: Figure data for fly distribution is limited by poor national sampling effort in relation to flies and taxonomic identification of species. We have presented the available data from the most populated national records database. If a species occurs in a region then the potential for widespread distribution throughout that region is high. We agree the location of horticultural areas is needed to understand the synchrony between fly and crop distribution and have included the description reference ‘Horticulture in Australia, occurs primarily within the coastal regions of Queensland, central and eastern New South Wales, southern Victoria, northern and south eastern Tasmania, southeast South Australia, southern Western Australia and the Northern Territory. Thus, there is considerable synchrony between potential fly pollinators and key crop distribution (Figure 2).’ Any potential for spatial mismatch can now be assessed by the reader as minimal given candidate species were selected in part based on their distribution alignment with key cropping regions as indicated in the review.

Page 14, the use of the word ‘native’ is inappropriate here and has been amended to ‘local’ as release of managed pollinators should be matched to species already present, native or otherwise.  
